# FILM: Following Instructions in Language with Modular Methods

**So Yeon Min**[1]  **Devendra Singh Chaplot**[2]  **Pradeep Ravikumar**[1]

**Yonatan Bisk**[1]  **Ruslan Salakhutdinov**[1]

[1] Carnegie Mellon University
{soyeonm, pradeepr, ybisk, rsalakhu}@cs.cmu.edu

[2] Facebook AI Research
dchaplot@fb.com

## Abstract

Recent methods for embodied instruction following are typically trained end-to-end using imitation learning. This often requires the use of expert trajectories and low-level language instructions. Such approaches assume that neural states will integrate multimodal semantics to perform state tracking, building spatial memory, exploration, and long-term planning. In contrast, we propose a modular method with structured representations that (1) builds a semantic map of the scene and (2) performs exploration with a semantic search policy, to achieve the natural language goal. Our modular method achieves SOTA performance (24.46%) with a substantial (8.17 % absolute) gap from previous work while using less data by eschewing both expert trajectories and low-level instructions. Leveraging low-level language, however, can further increase our performance (26.49%).[1] Our findings suggest that an explicit spatial memory and a semantic search policy can provide a stronger and more general representation for state-tracking and guidance, even in the absence of expert trajectories or low-level instructions.[2]

## 1 Introduction

Human intelligence simultaneously processes data of multiple modalities, including but not limited to natural language and egocentric vision, in an embodied environment. Powered by the success of machine learning models in individual modalities (Devlin et al., 2018; He et al., 2016; Voulodimos et al.; Anderson et al., 2018a), there has been growing interest to build multimodal embodied agents that perform complex tasks. An incipient pursuit of such interest was to solve the task of Vision Language Navigation (VLN), for which the agent is required to navigate to the goal area given a language instruction (Anderson et al., 2018b; Fried et al., 2018; Zhu et al., 2020).

Embodied instruction following (EIF) presents a more complex and human-like setting than VLN or Object Goal Navigation (Gupta et al., 2017; Chaplot et al., 2020b; Du et al., 2021); beyond just navigation, agents are required to execute sequences of sub-tasks that entail both navigation and interaction actions from a language instruction (Fig. 1). The additional challenges posed by EIF are threefold - the agent has to understand compositional instructions of multiple types and subtasks, choose actions from a large action space and execute them for longer horizons, and localize objects in a fine-grained manner for interaction (Nguyen et al., 2021).

Most existing methods (Zhang & Chai, 2021; Kim et al., 2021; Nottingham et al., 2021) for EIF have relied on neural memory of various types (transformer embeddings, LSTM state), trained end-to-end with expert trajectories upon raw or pre-processed language/visual inputs. However, EIF remains a very challenging task for end-to-end methods as they require the neural net to simultaneously learn state-tracking, building spatial memory, exploration, long-term planning, and low-level control.

In this work, we propose **FILM** ✍ (Following Instructions in Language with Modular methods). FILM consists of several modular components that each (1) processes language instructions into

---

[1] The official ALFRED leaderboard: https://leaderboard.allenai.org/alfred/submissions/public.
[2] Project webpage with code and pre-trained models: https://soyeonm.github.io/FILM_webpage/

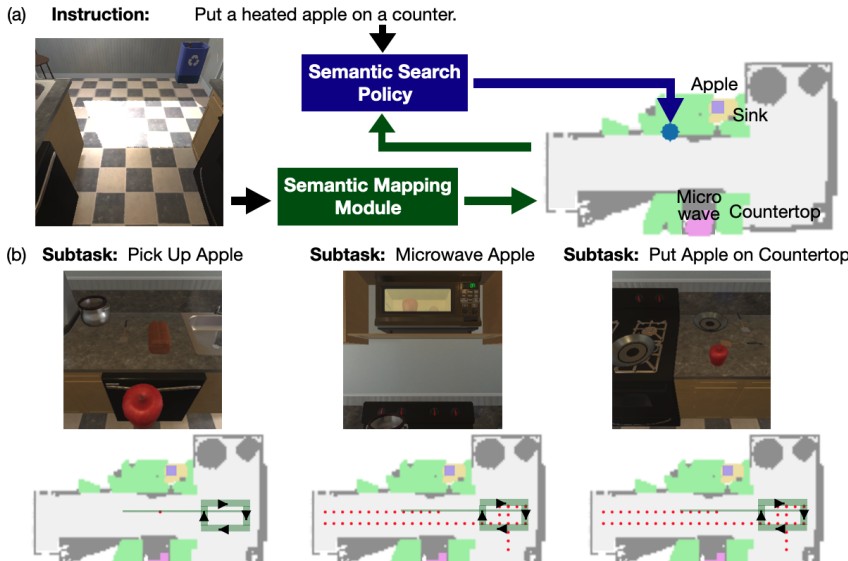

**Figure 1:** An Embodied Instruction Following (EIF) task consists of multiple subtasks. (a) **FILM method overview**: The agent receives the language instruction and the egocentric vision of the frame. At every time step, a semantic top-down map of the scene is updated from predicted depth and instance segmentation. Until the subgoal object is observed, a search goal (blue dot) is sampled from the semantic search policy. (b) **Example trajectories**: Trajectory of an existing model (HiTUT (Zhang & Chai, 2021)) is plotted in a straight green line, and that of FILM is in dotted red. While HiTUT's agent travels repeatedly over a path of closed loop (thick green line, arrow pointing in the direction of travel), FILM's semantic search allows better exploration and the agent sufficiently explores the environment and completes all subtasks.

structured forms (*Language Processing*), (2) converts egocentric visual input into a semantic metric map (*Semantic Mapping*), (3) predicts a search goal location (*Semantic Search Policy*), and (4) outputs subsequent navigation/ interaction actions (*Deterministic Policy*). FILM overcomes some of the shortcomings of previous methods by leveraging a modular design with structured spatial components. Unlike many of the existing methods for EIF, FILM does ***not*** require any input that provides sequential guidance, namely expert trajectories or low-level language instructions. While Blukis et al. (2021) recently introduced a method that uses a structured spatial memory, it comes with some limitations from the lack of explicit semantic search and the reliance on expert trajectories.

On the ALFRED (Shridhar et al., 2020) benchmark, FILM achieves State-of-the-Art performance (24.46%) with a large margin (8% absolute) from the previous SOTA (Blukis et al., 2021). Most approaches rely on low-level instructions, and we too find that including them leads to an additional 2% improvement in success rate (26.49%). FILM's strong performance and our analysis indicate that an explicit structured spatial memory coupled with a semantic search policy can provide better state-tracking and exploration, even in the absence of expert trajectories or low-level instructions.

## 2 RELATED WORK

A plethora of works have been published on embodied vision and language tasks, such as VLN (Anderson et al., 2018b; Fried et al., 2018; Zhu et al., 2020), Embodied Question Answering (Das et al., 2018; Gordon et al., 2018), and topics of multimodal representation learning (Wang et al., 2020; Bisk et al., 2020), such as Embodied Language Grounding (Prabhudesai et al., 2020). For Visual Language Navigation, which is the most comparable to the setting of our work, methods with impressive performances (Ke et al., 2019; Wang et al., 2019; Ma et al., 2019) have been proposed since the introduction of R2R (Anderson et al., 2018b). While far from conquering VLN, these methods have shown up to 61% success rate on unseen test environments (Ke et al., 2019).

For the more challenging task of Embodied Instruction Following (EIF), multiple methods have been proposed with differing levels of modularity in the model structure. As a baseline, Shridhar et al. (2020) has presented a Seq2Seq model with an attention mechanism and a progress monitor, while Pashevich et al. (2021) proposed to replace to seq2seq model with an episodic transformer. These methods take the concatenation of language features, visual features, and past trajectories as input and predict the subsequent action end-to-end. On the other hand, Kim et al. (2021); Zhang

& Chai (2021); Nguyen et al. (2021) modularly process raw language and visual inputs into structured forms, while keeping a separate "action prediction module" that outputs low-level actions given processed language outputs. Their "action taking module" itself is trained end-to-end and relies on neural memory that "implicitly" tracks all of spatial, progressive, and states of the agent. Unlike these methods, FILM's structured language/ spatial representations make reasons for failure transparent and elucidates directions to improve individual components.

Recently, Blukis et al. (2021) has proposed a more modular method with a persistent and structured spatial memory. Language and visual input are transformed into respectively high-level actions and the 3D map. With the 3D map and high-level actions as input, low-level actions are predicted with a value-iteration network (VIN). Navigation goals for the VIN are sampled from a model trained on interaction pose labels from expert trajectories. Among all proposed methods for EIF, FILM necessitates the least information (neither low-level instructions nor expert trajectories are needed, although the former can be taken as an additional input). Furthermore, FILM addresses the problem of search/ exploration of goal objects.

Various works in visual navigation with semantic mapping are also relevant. Simultaneous Localization and Mapping (SLAM) methods, which build 2D or 3D obstacle maps, have been widely used (Fuentes-Pacheco et al., 2015; Izadi et al., 2011; Snavely et al., 2008). In contrast to these works, recent methods (Chaplot et al., 2020b;a) build semantic maps with differentiable projection operations, which restrain egocentric prediction errors amplifying in the map. The task of Chaplot et al. (2020b;a) is object goal navigation, a much simpler task compared to EIF. Furthermore, while Chaplot et al. (2020b) employs a semantic exploration policy, our and their semantic policies serve fundamentally different purposes; while their policy guides a general sense of direction among multiple rooms in the search for large objects (e.g. fridge), ours guides the search for potential locations of small and flat objects which have little chance of detection at a distance. Also, our semantic policy is conditioned on language instructions. Blukis et al. (2018a;b) also successfully utilized semantic 2D maps in grounded language navigation tasks. These works are for quadcopters, whose fields of view almost entirely cover the scene and the need for "search" or "exploration" is less crucial than for pedestrian agents. Moreover, their settings only involve navigation with a single subtask.

## 3 TASK EXPLANATION

We utilize the ALFRED benchmark. The agent has to complete household tasks given only natural language instructions and egocentric vision (Fig. 1). For example, the instruction may be given as "Put a heated apple on the counter," with low-level instructions (which FILM does not use by default) further explaining step-by-step lower level actions. In this case, one way to "succeed" in this episode is to sequentially (1) pick up the apple, (2) put the apple in the microwave, (3) toggle the microwave on/off, (4) pick up the apple again, and (4) place it on the countertop. Episodes run for a significantly longer number of steps compared to benchmarks with only single subgoals; even expert trajectories, which are maximally efficient and perform only the strictly necessary actions (without any steps to search for an object), are often longer than 70 steps.

There are seven types of tasks (Appendix A.1), from relatively simple types (e.g. Pick & Place) to more complex ones (e.g. Heat & Place). Furthermore, the instruction may require that an object is "sliced" (e.g. Slice bread, cook it in the microwave, put it on the counter). An episode is deemed "success" if the agent completes all sub-tasks within 10 failed low-level actions and 1000 max steps.

## 4 METHODS

FILM consists of three learned modules: (1) Language Processing (LP), (2) Semantic Mapping, and (3) Semantic Search Policy; and one purely deterministic navigation/ interaction policy module (Fig. 2). At the start of an episode, the LP module processes the language instruction into a sequence of subtasks. Every time step, the semantic mapping module receives the egocentric RGB frame and updates the semantic map. If the goal object of the current subtask is not yet observed, the semantic search policy predicts a "search goal" at a coarse time scale; until the next search goal is predicted, the agent navigates to the current search goal with the deterministic policy. If the goal is observed, the deterministic policy decides low-level controls for interaction actions (e.g. "Pick Up" object).

### 4.1 LANGUAGE PROCESSING (LP)

The language processing (LP) module transforms high-level instructions into a structured sequence of subtasks (Fig. 3). It consists of two BERT (Devlin et al., 2018) submodules that receive the in-

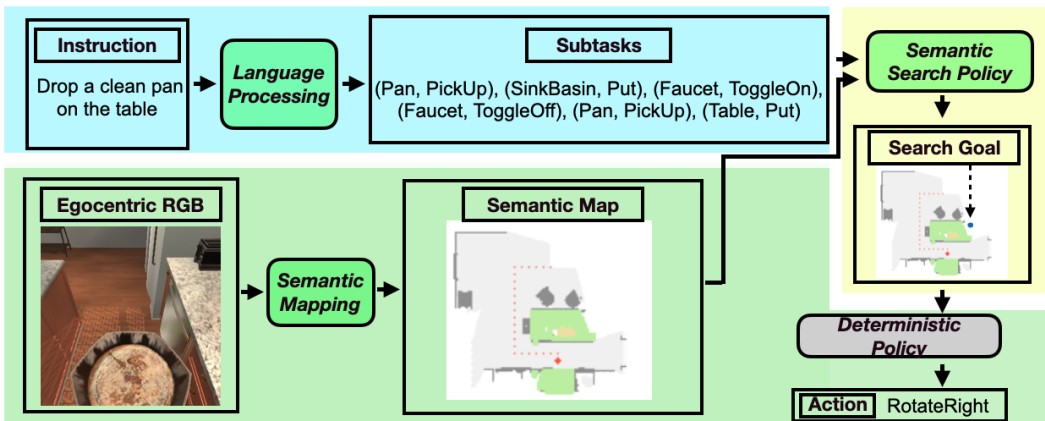

**Figure 2: FILM method overview**. The "grouping" in blue, green, and yellow denote the coarseness of time scale (blue: at the beginning of the episode, green: at every time step, yellow: at a coarser time scale of every 25 steps). At the beginning of the episode, the Language Processing module processes the instruction into subtasks. At every time step, Semantic Mapping converts egocentric into RGB a top-down semantic map. The semantic search policy outputs the search goal at a coarse time scale. Finally, the Deterministic Policy decides the next action. Modules in bright green are learned; the deterministic policy (grey) is not.

struction as an input at the beginning of the episode. The first submodule (*BERT type classification*) receives the instruction and predicts the "type" of the instruction - one of the seven types stated in Appendix A.1. The second submodule (*BERT argument classification*) receives both the instruction and the predicted type as input and predicts the "arguments" - (1) *"obj"* for the object to be picked up, (2) *"recep"* for the receptacle where *"obj"* should be ultimately placed, (3) *"sliced"* for whether *"obj"* should be sliced, and (4) *"parent"* for tasks with intermediate movable receptacles (e.g. "cup" in "Put a knife in a cup on the table" of Appendix A.1). An object in ALFRED is always an instance of either *"obj"* or *"recep"*; *"parent"* objects are a subset of *"recep"* objects that are movable. We train a separate BERT model for each argument predictor. The two submodules are easily trainable with supervised learning since the type and the four arguments are provided in the training set. Models use only the CLS token for classification, and they do not share parameters; all layers of "bert-base-uncased" were fine-tuned.

Due to the patterned nature of instructions, we can match the predicted "type" of the instruction to a "type template" with blank arguments. For example, if the instruction is classified as the "clean & place" type, it is matched to the template "(*Obj*, PickUp), (SinkBasin, Put), (Faucet, ToggleOn), (Faucet, ToggleOff), (*Obj*, PickUp), (*Recep*, Put)". If the "sliced" argument is predicted to be true from argument classification, subtasks of "(Knife, PickUp), (*Obj*, Slice), (Sink, PutObject)" will be added at the beginning of the template (with the (Sink, PutObject) to make the agent drop the knife). Filling in the "type template" with predictions of the second model, we obtain a list of subtasks (bottom of Fig. 3b) to be completed in the current episode. The "type templates" were designed by hand in less than 20 minutes. In section 5.2, we discuss the effect of using a LP module without the template assumption, for fair comparison with other works. Appendix A.9 contains more details.

## 4.2 Semantic Mapping Module

We designed the semantic mapping module (Appendix A.2) with inspirations from prior work (Chaplot et al., 2020b). Egocentric RGB is first processed into depth map and instance segmentation, with MaskRCNN (He et al., 2017) (and its implementation by Shridhar et al. (2021)) and the depth prediction method of Blukis et al. (2021); details of the training are explained in Section 5 [3]. These pre-trained, off-the-shelf models were finetuned on the training scenes of ALFRED. Once processed, the depth observation is transformed to a point cloud, of which each point is associated with the predicted semantic categories. Finally, the point cloud is binned into a voxel representation; this summed over height is the semantic map. The map is locally updated and aggregated over time.

The resulting semantic map is an allocentric $(C + 2) \times M \times M$ binary grid, where $C$ is the number of object categories and each of the $M \times M$ cells represents a 5cm $\times$ 5cm space of the scene. The $C$ channels each represent whether a particular object of interest was observed; the two extra channels denote whether obstacle exists and whether exploration happened

---

[3]We use the publicly released code of Shridhar et al. (2021); Blukis et al. (2021).

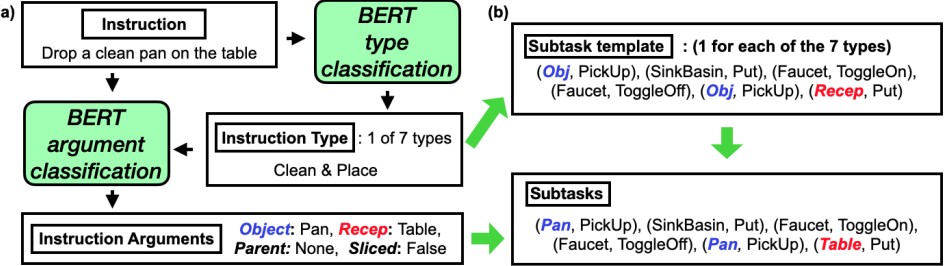

**Figure 3: The Language Processing module.** (a): Two BERT models respectively predict the "type" and the "arguments" of the instruction. (b): The predicted "type" from (a) is matched with a template, and the "arguments" of the template is filled with the predicted "argument."

in a particular 5cm $\times$ 5cm space. Thus, the $C + 2$ channels are a semantic/spatial summary of the corresponding space. We use $M = 240$ (12 meters in the physical world) and $C = 28 +$ (number of additional subgoal objects in the current task). "28" is the number of "receptacle" objects (e.g. "Table", "Bathtub"), which are usually large and easily detected; in the example of Fig. 1, there is one additional subgoal object ("Apple"). Please see Appendix A.2 on details of the dynamic handling of $C$.

### 4.3 SEMANTIC SEARCH POLICY

The semantic search policy outputs a coarse 2D distribution for potential locations of a small subgoal object (Fig. 4), given a semantic map with the 28 receptacle objects only (e.g. "Countertop", "Shelf"). The discovery of a small object is difficult in ALFRED due to three reasons - (1) many objects are tiny (some instances of "pencil" occupies less than 200 pixels even at a very close view), (2) the field of view is small due to the camera horizon mostly being downward[4], (3) semantic segmentation, despite being fine-tuned, cannot detect small objects at certain angles. The role of the semantic search policy is to predict search locations for small objects, upon the observed spatial configuration of larger ones. While existing works surmise the "implicit" learning of search locations from expert trajectories, we directly learn an explicit policy without expert trajectories.

The policy is trained via supervised learning. For data collection, we deploy the agent without the policy in the training set and gather the (1) semantic map with only receptacle objects and (2) the ground truth location of the subgoal object after every 25 steps. A model of 15 layers of CNN with max-pooling in between (details in Appendix A.3) outputs an $N \times N$ grid, where $N$ is smaller than the original map size $M$; this is a 2D distribution for the potential location of the subgoal object. Finally, the KL divergence between this and a pseudo-ground truth "coarse" distribution whose mass is uniformly distributed over all cells with the true location of the subgoal object is minimized ($\min_p KL(p||q)$ where $p$ is the coarse ground truth and $q$ is the coarse prediction). At deployment, the "search goal" is sampled from the predicted distribution, resized to match the original map size of $M \times M$ (e.g. $240 \times 240$), with mass in the coarse $N \times N$ (e.g. $8 \times 8$) grid uniformly spread out to the $\lfloor \frac{M}{N} \rfloor \times \lfloor \frac{M}{N} \rfloor$ area centered on it. Because arriving at the search goal requires time, the policy operates at a "coarse" time scale of 25 steps; the agent navigates towards the current search goal until the next goal is sampled or the subgoal object is found (more details in Section 4.4).

Fig. 4 shows a visualization of the semantic search policy's outputs. The policy provides a reasonably close estimate of the true distribution; the predicted mass of "bowl" is shared around observed furniture that it can appear on, and that of "faucet" peaks around the sink/ the end of the bathtub. While we chose $N = 8$ as the grid size, Appendix A.4 provides a general bound for choosing $N$.

### 4.4 DETERMINISTIC POLICY

Given (1) the predicted subtasks, (2) the most recent semantic map, and (3) the search goal sampled at a coarse time scale, the deterministic policy outputs a navigation or interaction action (Fig. 2).

Let $[(obj_1, action_1), ... , (obj_k, action_k)]$ be the list of subtasks and the current subtask be $(obj_i, action_i)$. If $obj_i$ is observed in the current semantic map, the closest $obj_i$ is selected as the goal; otherwise, the sample from the semantic search policy is chosen as the goal (Section 4.3). The agent then navigates towards the goal via the Fast Marching Method (Sethian, 1996) and performs

---

[4]The agent mostly looks down $45°$ in FILM for correct depth prediction. Looking down is common in existing models as well (Kim et al., 2021; Zhang & Chai, 2021; Blukis et al., 2021).

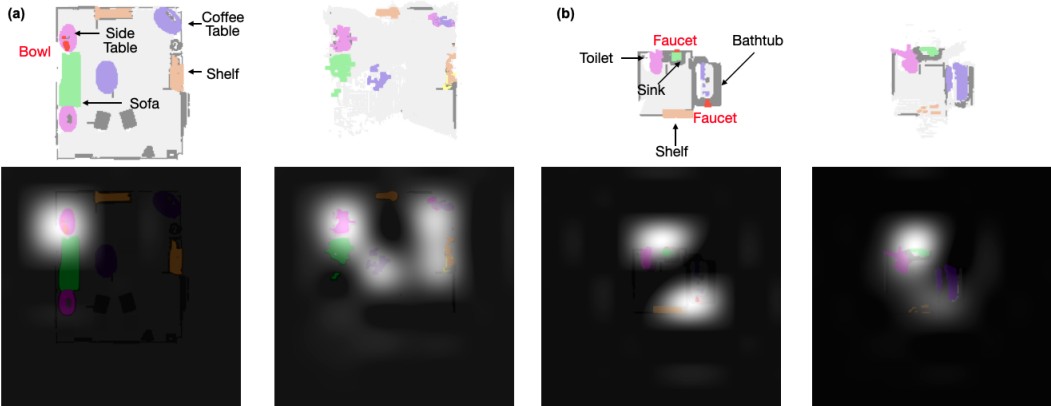

**Figure 4: Example visualization of semantic search policy outputs.** In each of (a), (b), Top left: map built from ground truth depth/ segmentation, Top right: map from learned depth/ segmentation, Bottom left: ground truth "coarse" distribution, Bottom right: predicted "coarse" distribution. (a): While the true location of the "bowl" was on the upper left coffee table, the policy distributes mass over all furniture likely to have it on. (b): The true location of the faucet is on the sink and at the end of the bathtub. While the policy puts more mass near the sink, it also allocates some to the end of the bathtub.

the required interaction actions. While this "low-level" policy could be learned with imitation or reinforcement learning, we used a deterministic one based on the findings of earlier work that observed that the Fast Marching Method performs as well as a learned local navigation policy (Chaplot et al., 2020b). When the agent successfully executes the required interaction $action_i$ (which can be determined by the change in the egocentric RGB), the pointer of subtasks is advanced to $i + 1$ or the task is completed. More details and pseudocode are provided in Appendix A.5.

## 5 EXPERIMENTS AND RESULTS

We explain the metrics, evaluation splits, and baselines against which FILM is compared. Furthermore, we describe training details of each of the learned components of FILM.

**Metrics** Success Rate (SR) is a binary indicator of whether all subtasks were completed. The goal-condition success (GC) of a model is the ratio of goal-conditions completed at the end of an episode. For example, in the example of Fig. 1, there are three goal-conditions - a pan must be "cleaned", a pan should rest on a countertop, and a "clean" pan should rest on a countertop. Both SR and GC can be weighted by (path length of the expert trajectory)/ (path length taken by the agent); these are called path length weighted SR (PLWSR) and path length weighted GC (PLWGC).

**Evaluation Splits** The test set consists of "Tests Seen" (1533 epsiodes) and "Tests unseen" (1529 episodes); the scenes of the latter entirely consist of rooms that do not appear in the training set, while those of the former only consist of scenes seen during training. Similarly, the validation set is partitioned into "Valid Seen" (820 epsiodes) and "Valid Unseen" (821 epsiodes). The official leaderboard ranks all entries by the SR on Tests Unseen.

**Baselines** There are two kinds of baselines: those that use low-level sequential instructions (Kim et al., 2021; Zhang & Chai, 2021; Nguyen et al., 2021; Pashevich et al., 2021) and those that do not (Nottingham et al., 2021; Blukis et al., 2021). While FILM does not necessitate low-level instructions, we report results with and without them and compare them against methods of both kinds.

**Training Details of Learned Components** In the LP module, BERT type classification and argument classification were trained with AdamW from the `Transformer` (Wolf et al., 2019) package; learning rates are 1e-6 for type classification and {1e-4,1e-5,5e-5,5e-5} for each of "object", "parent", "recep", "sliced" argument classification. In the Semantic Mapping module, separate depth models for camera horizons of 45° and 0° were fine-tuned from an existing model of HLSM (Blukis et al., 2021), both with learning rate 1e-3 and the AdamW optimizer (epsilon 1e-6, weight decay 1e-2). Similarly, separate instance segmentation models for small and large objects were fine-tuned, starting from their respective parameters released by Shridhar et al. (2021), with learning rate 1e-3 and the SGD optimizer (momentum 0.9, weight decay 5e-4). Finally, the semantic search policy was trained with learning rate 5e-4 and the AdamW optimizer (epsilon 1e-6). Appendix A.2 and A.3 discuss more details on the architectures of semantic mapping/ semantic search policy modules. The readme of our code states protocols and commands so that readers can reproduce all expriments.

**Table 1:** Test results. Top section uses step-by-step instructions; bottom section does not. **Bold** numbers are top scores in each section. Blue numbers are the top SR on Tests Unseen (by which the leaderboard is ranked).

| Method | | Tests Seen | | | | Tests Unseen | | | |
|---|---|---|---|---|---|---|---|---|---|
| | | PLWGC | GC | PLWSR | SR | PLWGC | GC | PLWSR | SR |
| **Low-level Sequential Instructions + High-level Goal Instruction** | | | | | | | | | |
| SEQ2SEQ | (Shridhar et al., 2020) | 6.27 | 9.42 | 2.02 | 3.98 | 4.26 | 7.03 | 0.08 | 3.9 |
| MOCA | (Singh et al., 2020) | 22.05 | 28.29 | 15.10 | 22.05 | 9.99 | 14.28 | 2.72 | 5.30 |
| E.T. | (Pashevich et al., 2021) | - | 36.47 | - | 28.77 | - | 15.01 | - | 5.04 |
| E.T. + synth. data | (Pashevich et al., 2021) | **34.93** | 45.44 | 27.78 | 38.42 | 11.46 | 18.56 | 4.10 | 8.57 |
| LWIT | (Nguyen et al., 2021) | 23.10 | 40.53 | **43.10** | 30.92 | 16.34 | 20.91 | 5.60 | 9.42 |
| HiTUT | (Zhang & Chai, 2021) | 17.41 | 29.97 | 11.10 | 21.27 | 11.51 | 20.31 | 5.86 | 13.87 |
| ABP | (Kim et al., 2021) | 4.92 | **51.13** | 3.88 | **44.55** | 2.22 | 24.76 | 1.08 | 15.43 |
| FILM W.O. SEMANTIC SEARCH | | 13.10 | 35.59 | 9.43 | 25.90 | 13.37 | 35.51 | 10.17 | 23.94 |
| FILM 🖺 | | 15.06 | 38.51 | 11.23 | 27.67 | **14.30** | **36.37** | **10.55** | **26.49** |
| **High-level Goal Instruction Only** | | | | | | | | | |
| LAV | (Nottingham et al., 2021) | 13.18 | 23.21 | 6.31 | 13.35 | 10.47 | 17.27 | 3.12 | 6.38 |
| HiTUT G-only | (Zhang & Chai, 2021) | - | 21.11 | - | 13.63 | - | 17.89 | - | 11.12 |
| HLSM | (Blukis et al., 2021) | 11.53 | 35.79 | 6.69 | 25.11 | 8.45 | 27.24 | 4.34 | 16.29 |
| FILM W.O. SEMANTIC SEARCH | | 12.22 | 34.41 | 8.65 | 24.72 | 12.69 | 34.00 | 9.44 | 22.56 |
| FILM 🖺 | | **14.17** | **36.15** | **10.39** | **25.77** | **13.13** | **34.75** | **9.67** | **24.46** |

## 5.1 RESULTS

Table 8 shows test results. FILM achieves state-of-the-art performance across both seen and unseen scenes in the setting where only high-level instructions are given. It achieves 8.17% absolute (50.15% relative) gain in SR on Tests Unseen, and 0.66% absolute (2.63% relative) gain in SR on Tests Seen over HLSM, the previous SOTA.

FILM performs competitively even compared to methods that require low-level step-by-step instructions. They can be used as additional inputs to the LP module, with the low-level instruction appended to the high-level instruction for both BERT type classification and BERT argument classification. Under this setting, FILM achieves 11.06% absolute (71.68% relative) gain in SR on Tests Unseen compared to ABP. Notably, FILM performs similarly across Tests Seen and Tests Unseen, which implies FILM's strong generalizability. This is in contrast to that methods that require low-level instructions, such as ABP, E.T., LWIT, MOCA, perform very well on Tests Seen but much less so on unseen scenes. In a Sim2Real situation, these methods will excel if the agent can be trained in the exact household it will be deployed in, with multiple low-level instructions and expert trajectories. In the more realistic and cost-efficient setting where the agent is trained in a centralized manner and has to generalize to new scenes, FILM will be more adequate.

It is also notable that the semantic search policy significantly increases not only SR and GC, but also their path-length weighted versions. On Tests Seen, the gap of PLWSR between FILM with/ without semantic search is larger than the corresponding gap of SR (for both with/ without low-level instructions). This suggests that the semantic policy boosts the efficiency of trajectories. The results in Appendix A.8 show that the improvement by the semantic policy is reproduced across multiple seeds; the protocols for reproduction are explained along with the result.

## 5.2 ABLATIONS STUDIES AND ERROR ANALYSIS

**Errors due to perception and language processing.** To understand the importance of FILM's individual modules, we consider ablations on the base method, the base method with low-level language, and with ground truth visual/ language inputs. Table 2 shows ablations on the development sets. While the improvement from gt depth is large in unseen scenes (10.64%), it is incremental on seen scenes (1.48%); on the other hand, gt segmentation significantly boosts performances in both cases (9.26% / 9.26%). Thus, among visual perception, segmentation is a bottleneck in both seen/ unseen scenes, and depth is a bottleneck only in the latter. On the other hand, while a large gain in SR comes from using ground truth language (7.43 % / 4.22 %), that from adding low-level language as input is rather incremental. We additionally analyze the effect of the template assumption (explained in the second paragraph of Section 4.1), by reporting the performance with a Language Processing module without this assumption. The results drop without the templates but not by a large margin. Appendix A.9 explains the details of this auxiliary Language Processing module.

**Error modes.** Table 3 shows common error modes of FILM; the metric is the percent of episodes that failed (in SR) from a particular error out of all failed episodes. The main failures in valid unseen scenes are due to failures in (1) locating the subgoal object (due to the small field of view, imperfect segmentation, ineffective exploration), (2) locating the subgoal object because it is in a closed re-

**Table 2:** Ablation results on validation splits. Base Method is FILM with semantic search policy.

| Method | Val Seen | | Val Unseen | |
|---|---|---|---|---|
| | GC | SR | GC | SR |
| Base Method | 37.20 | 24.63 | 32.45 | 20.10 |
| + low-level language | 38.54 | 25.24 | 32.89 | 20.61 |
| + gt seg. | 45.46 | 34.02 | 42.88 | 29.35 |
| + gt depth | 38.21 | 26.59 | 42.91 | 30.73 |
| + gt depth, gt seg. | 55.54 | 43.22 | 64.31 | 55.05 |
| + gt depth, gt seg., gt lang. | 59.47 | 47.44 | 69.13 | 62.48 |
| - template assumption | 31.46 | 20.37 | 31.14 | 18.03 |

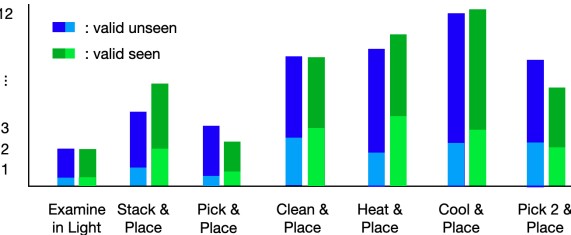

**Figure 5:** Average number of subtasks *completed until failure*, by task type (light green/ light blue respectively for valid seen/ unseen). Dark green/ blue: average number of *total* subtasks in valid seen/ unseen.

**Table 3: Error Modes**. Table showing percentage of errors due to each failure mode for FILM on the Val set.

| Error mode | Seen | Unseen |
|---|---|---|
| Goal object not found | 23.30 | 26.07 |
| Interaction failures | 6.96 | 8.54 |
| Collisions | 6.96 | 11.00 |
| Object in closed receptacle | 18.44 | 16.16 |
| Language processing error | 18.53 | 24.54 |
| Others | 25.81 | 13.69 |

**Table 4:** Performance by task type of base model on validation.

| Task Type | Val Seen | | Val Unseen | |
|---|---|---|---|---|
| | GC | SR | GC | SR |
| Overall | 37.20 | 24.63 | 32.45 | 20.10 |
| Examine | 50.00 | 34.41 | 45.06 | 29.65 |
| Pick & Place | 27.46 | 26.92 | 16.67 | 16.03 |
| Stack & Place | 23.74 | 10.71 | 9.90 | 1.98 |
| Clean & Place | 58.56 | 44.04 | 48.89 | 33.63 |
| Cool & Place | 27.04 | 12.61 | 27.41 | 14.04 |
| Heat & Place | 40.21 | 22.02 | 37.77 | 23.02 |
| Pick 2 & Place | 40.37 | 23.77 | 29.28 | 11.84 |

ceptacle (cabinet, drawer, etc), (3) interaction (due to object being too far or not in field of view, bad segmentation mask), (4) navigation (collisions), (5) correctly processing language instructions, (6) others, such as the deterministic policy repeating a loop of actions from depth/ segmentation failures and 10 failed actions accruing from a mixture of different errors. A failed episode is classified to the error type that occurs "earlier" (e.g. If the subtasks were processed incorrectly and also there were 10 consecutive collisions, this episode is classified as (5) (failure in incorrectly processsing language instructions) since the LP module comes "earlier" than the collisions). More details are in Appendix A.6. As seen in Table 3, *goal object not found* is the most common error mode, typically due to objects being small and not visible from a distance or certain viewpoints. Results of the next subsection show that this error is alleviated by the semantic search policy in certain cases.

**Performance over different task types.** To understand FILM's strengths/ weaknesses across different types of tasks, we further ablate validation results by task type in Table 4. Figure 5 shows the average number of subtasks completed for failed episodes, by task type. First, the SR and GC for "Stack & Place" is remarkably low. Second, the number of the subtasks entailed with the task type does not strongly correlate with performance. While "Heat & Place" usually involves three more subtasks than "Pick & Place", the metrics for the former are much higher than those of the latter. Since task types inevitably occur in different kinds of scenes (e.g. "Heat & Place" only occurs in kitchens) and therefore involve different kinds of objects (e.g. "Heat & Place" involves food only), the results suggest that the success of the first PickUp action largely depends on the kinds of the scene and size and type of the subgoal objects rather than number of subtasks.

While the above error analysis is specific to FILM, its implications regarding visual perception may generally represent the weaknesses of existing methods for EIF, since most recent methods (ABP, HLSM, HiTUT, LWIT, E.T.) use the same family of segmentation/ detection models as FILM, such as Mask-RCNN and Fast-RCNN (Wang et al., 2017). Specifically, it could be that the inability to find a subgoal object is a major failure mode in the mentioned existing methods as well. On the other hand, FILM is not designed to search inside closed receptacles (e.g. cabinets), although subgoal objects dwell in receptacles quite frequently (Table 3); a future work to extend FILM should learn to perform a more active search.

## 5.3 Effects of the Semantic Search Policy

With Valid Unseen as the development set, we observed that the semantic search policy significantly helps to find small objects (Table 5); we use the percent of episodes in which the first goal object was found (%1st Goal Found) as a proxy, since it can be picked up (e.g. "Apple", "Pen")

**Table 5:** Dev set results (*valid unseen*) of FILM with/ without semantic search policy.

| Method | % 1st Goal Found | SR |
|---|---|---|
| HLSM (Blukis et al., 2021) | N/A | 11.8 |
| FILM with Search | **80.51** | **20.09** |
| FILM w.o. Search | 76.12 | 19.85 |

**Instruction: Put a large clean knife on the counter**

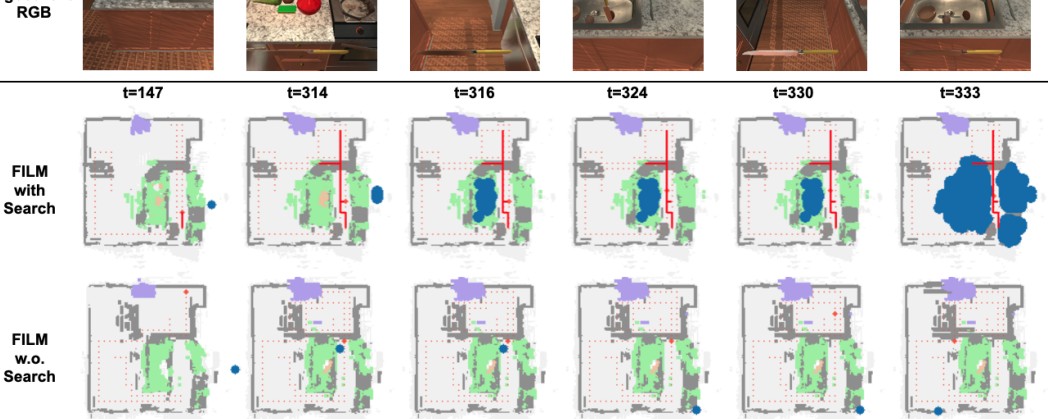

Figure 6: **Example trajectories of FILM with and without semantic search policy.** Paths near the subgoals that were traveled 3 times or more are in straight red. The goal (which can be the search goal or an observed instance of a subgoal object) is in blue.

Table 6: Performance with and without semantic search policy, by room size.

| Room Size | Small | | Large | |
|---|---|---|---|---|
| | FILM | FILM w.o. Search | FILM | FILM w.o. Search |
| SR | 26.70 | 26.63 | 15.17 | 14.74 |
| % 1st Goal Found | 79.32 | 81.02 | 80.13 | 73.72 |

Table 7: Performance with and without semantic search policy, by task type.

| Task Type | Clean & Place | | Other Types | |
|---|---|---|---|---|
| | FILM | FILM w.o. Search | FILM | FILM w.o. Search |
| SR | 33.63 | 14.16 | 17.94 | 20.16 |
| % 1st Goal Found | 87.61 | 79.65 | 79.38 | 75.56 |
| % 1st Recep Found | 80.53 | 69.03 | 58.05 | 55.93 |

and thus is usually small. Thus, we use FILM with semantic search as the "base method" (default) for all experiments/ ablations.

To further analyze when the semantic search policy especially helps, we ablate on room sizes and task types. Table 6 shows the SR and %1st Goal Found with and without search, by room size (details on the assignment of Room Size are in Appendix A.7). As expected, the semantic policy increases both metrics, especially so in large scenes. This is desirable since the policy makes the agent less disoriented in difficult scenarios (large scenes); the model without it is more susceptible to failing even the first subtask. Figure 6 is consistent with the trend of Table 6; it shows example trajectories of FILM with and without the semantic search policy in a large kitchen scene. Since the countertop appears in the bottom right quadrant of the map, it is desirable that the agent travels there to search for a "knife". While FILM travels to this area frequently (straight red line in Fig.6), FILM without semantic search mostly wanders in irrelevant locations (e.g. the bottom left quadrant).

Table 7 further shows the performance with and without search by task type. Notably, the gap of performance for the "clean & place" type is very large. In the large kitchen scene of "Valid Unseen" (Fig. 6), the "Sink" looks very flat from a distance and is hardly detected. The semantic policy induces the agent to travel near the countertop area and improves the localization of the 1st Recep ("Sink") for the "clean & place" type (Table 7). In conclusion, the semantic policy improves the localization of small and flat objects in large scenes.

## 6 CONCLUSION

We proposed FILM, a new modular method for embodied instruction following which (1) processes language instructions into structured forms (*Language Processing*), (2) converts egocentric vision into a semantic metric map (*Semantic Mapping*), (3) predicts a likely goal location (*Semantic Search Policy*), and (4) outputs subsequent navigation/ interaction actions (*Algorithmic Planning*). FILM achieves the state of the art on the ALFRED benchmark without any sequential supervision.

ETHICS STATEMENT

This research is for building autonomous agents. While we do not perform any experiments with humans, practitioners may attempt to extend and apply this technology in environments with humans. Such potential applications of this research should take privacy concerns into consideration.

All learned models in this research were trained using Ai2Thor (Kolve et al., 2019). Thus, they may be biased towards North American homes.

REPRODUCIBILITY STATEMENT

We thoroughly explain training details and model architectures in Section 5.1 and Appendix A.2, A.3. Project webpage with code, pre-trained models, and protocols to reproduce results is released here: https://soyeonm.github.io/FILM_webpage/.

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

# A    APPENDIX

## A.1    TASK DEFINITION

High and low-level instructions are both available to agents. There are 7 types of tasks (Fig 7. b) and the sequence of subtasks is templated according to the task type.

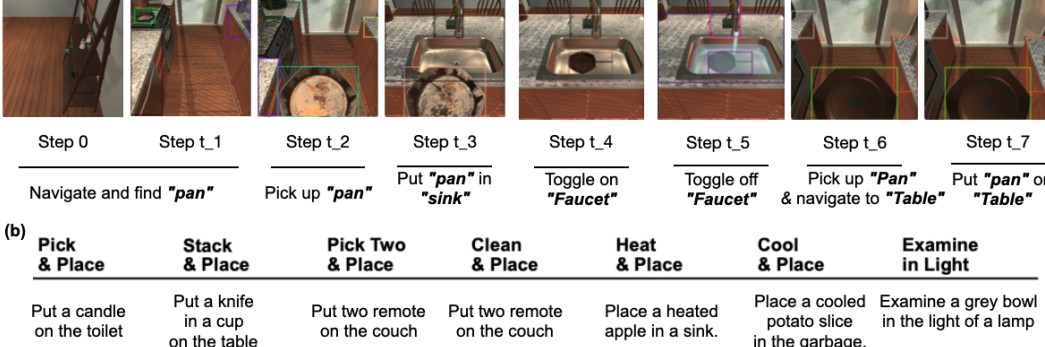

**Figure 7: ALFRED overview.** The goal is given in high level and low level language instructions. For and agent to achieve "success" of the goal, it needs to complete a sequence of interactions (as in the explanations in the bottom of the figure) and the entailed navigation between interactions.

## A.2    SEMANTIC MAPPING MODULE

Figure 8 is an illustration of the semantic mapping module. A depth map and instance segmentation is predicted from Egocentric RGB. Then the first and the later are respectively transformed into a point cloud and a semantic label of each point in the cloud, together producing voxels. The voxels are summed across height to produce the semantic map. Partial maps obtained at particular time steps are aggregated to the global map simply via "sum/ logical or."

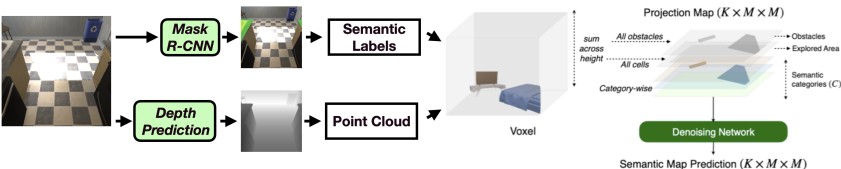

**Figure 8: Semantic mapping module.** Figure was partially taken from Chaplot et al. (2020b)

We dynamically control the number of objects $C$ for efficiency (because there are more than 100 objects in total). All receptacle objects (for input to the semantic policy) and all non-receptacle objects that appear in the subtasks are counted in $C$. For example, in an episode with the subtask [(Pan, PickUp), (SinkBasin, Put), (Faucet, ToggleOn), (Faucet, ToggleOff), (Pan, PickUp), (Table, Put)], all receptacle objects and "Pan", "Faucet" will be the $C$ objects indicated on the map.

## A.3    SEMANTIC SEARCH POLICY MODULE

The map from the previous subsection is passed into 7 layers of convolutional nets, each with kernel size 3 and stride 1. There is maxpooling between any two conv nets, and after the last layer, there is softmax over the 64 ($8 \times 8$) categories, for each of the $C_o$ (73) channels.

At deployment/ validation, if the agent is currently searching for the $c$th object, then a search location is sampled from the $c$th channel of the outputted $8 \times 8 \times C_o$ grid.

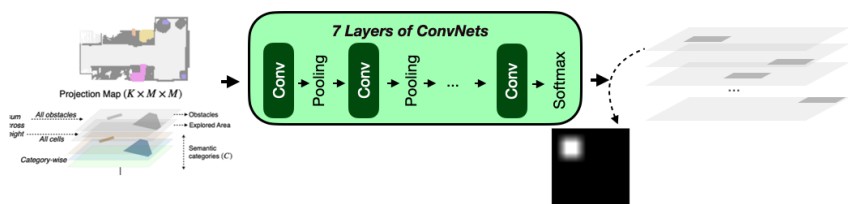

**Figure 9: Semantic search policy.**

## A.4 IMPACT OF GRID SIZE ON THE EFFECTIVENESS OF THE SEMANTIC SEARCH POLICY

While we chose $N = 8, \lfloor \frac{M}{N} \rfloor = 30$ for the size of the "coarse" cell of the semantic search policy, the desirable choice of $N$ may be different if a practitioner attempts to transfer FILM to different scenes/ tasks. While a "too fine" semantic policy will be hard to train due to sparseness of labels, a "too coarse" one will spread the mass of the distribution to widely.

Let us examine the "coarse" and "actual" ground truth distributions just in one direction (e.g. the horizontal direction). Let $F_X(x), C_X(x)$ be the "actual" and "coarse" ground truth CDFs in the horizontal direction. Also, let $L = \lfloor \frac{M}{N} \rfloor$ If the goal object occurs "$k$" times in the horizontal direction, then,

$$\sup_x |F_X(x) - C_X(x)| \le \frac{1}{k}(1 - \frac{1}{L}).$$

A similar result holds in the vertical direction. The bound above suggests that if the goal object occurs more frequently (smaller $\frac{1}{k}$), then a coarser $L$ (larger $1 - \frac{1}{L}$) is tolerable. On the other hand, if the goal object occurs very infrequently (larger $\frac{1}{k}$), then a coarse $M$ (larger $1 - \frac{1}{L}$) will result in $F_X$ and $C_X$ becoming too different in the worst case. Thus, it is desirable that practitioners choose $L$ (and in turn, $N$) based on the frequency of their goal objects, on average. Furthermore, a search policy with adaptive grid sizing should be explored as future work.

## A.5 PSEUDOCODE FOR THE DETERMINISTIC POLICY

Following the discussion of Section 4.4, let $[(obj_1, action_1), \dots , (obj_k, action_k)]$ be the list of subtasks, where the current subtask is $(obj_i, action_i)$. If $obj_i$ is observed in the current semantic map, the closest $obj_i$ is selected as the goal to navigate; otherwise, the sample from the semantic search policy is chosen as the goal (Section 4.3). The agent then navigates towards the closest $obj_i$ via the Fast Marching Method (Sethian, 1996). Once the stop distance is reached, the agent rotates 8 times to the left (at camera horizon 0, 45, 90,...) until $obj_i$ is detected in egocentric vision. Once $obj_i$ is in the current frame, the agents decides to take $action_i$ if two criteria are met: whether $obj_i$ is in the "center" of the frame, or whether the minimum depth towards $obj_i$ is in visibility distance of 1.5 meters). Otherwise, the agent "sidesteps" to keep $obj_i$ in the center frame or continue rotating to the left with horizon 0/45 until $obj_i$ is seen within visibility distance. If the agent executes $action_i$ and fails, the agent "moves backwards" and the map gets updated.

Below, we present a pseudocode for the deterministic navigation/ interaction policy. We first present explanations of some terms.

- "visible" means that an object is in the current RGB frame, and minimum (predicted) depth from the agent to it is less than or equal to 1.5 meters (which is set by ALFRED).

- "FMM" is Fast Marching Method (Sethian, 1996).

- We assume that a new RGB frame is given as $time\_step \leftarrow time\_step + 1$

- MoveBehind, SideStep, RotateBack are not actions in ALFRED; they are defined by us.
     MoveBehind - RotateRight, MoveAhead, RotateLeft
     SideStep - RotateRight/Left, MoveAhead, RotateLeft/Right
     RotateBack - RotateRight, RotateRight

---

**Algorithm 1** Navigation/ interaction algorithm in an episode

---

1: **Input:** List of goal tuples - $[(obj_1, action_1), ... , (obj_k, action_k)]$
2: **Output:** Task Success - True/False
3:
4: $timestep \leftarrow 1$
5: $goal\_pointer \leftarrow 1$
6: Sample $g$ from the semantic search policy
7: $execute\_interaction \leftarrow False$
8: $stop \leftarrow False$
9: $subtask\_success \leftarrow False$
10: $move\_pointer \leftarrow 0$
11: $task\_success \leftarrow False$
12:
13: $obj_i \leftarrow obj_{goal\_pointer}$; $action_i \leftarrow action_{goal\_pointer}$
14:
15: **while** $goal\_pointer \leq k$ **do**
16:     **while** $timestep \leq 1000$ **do**
17:         update semantic map
18:
19:         **if** $stop$ **then**
20:             **if** $execute\_interaction$ **then**
21:                 Execute $action_i$
22:                 **if** $action_i$ done successfully **then**
23:                     $subtask\_success \leftarrow True$
24:             **else**
25:                 **if** $obj_i$ visible in current frame and $obj_i$ in the center of the frame **then**
26:                     $execute\_interaction \leftarrow False$
27:                     Execute LookDown $0°$                             ▷ void action
28:                 **else**
29:                     **if** previous action was OpenObject or CloseObject and not $subtask\_success$ **then**
30:                         Execute MoveBehind
31:                     **else if** previous action was PutObject and not $subtask\_success$ **then**
32:                         Re-dilate $g$ in the semantic map
33:                         Execute RotateBack
34:                     **else if** $obj_i$ visible but not in center of the frame **then**
35:                         Execute SideStep
36:                     **else**                 ▷ Rotate with camera horizons $0°$, $45°$ until $obj_i$ is visible
37:                       **if** $move\_pointer < 4$ **then**
38:                         Execute RotateLeft
39:                       **else**
40:                         **if** $move\_pointer == 4$ **then**
41:                           Execute LookDown $45°$
42:                         Execute RotateLeft
43:                     $move\_pointer \leftarrow move\_pointer + 1$ (mod 8)
44:
45:         **else**
46:             **if** not $(obj_i$ found) **then**
47:                 Execute one of (RotateLeft, RotateRight, MoveAhead) with FMM to $g$
48:             **else**
49:                 $g \leftarrow$ closest $obj_i$ in the semantic map
50:                 **while** distance to $g \geq 0.65$ meters **do**
51:                     Execute one of (RotateLeft, RotateRight, MoveAhead) with FMM to $g$
52:                 **if** distance to $g < 0.65$ meters **then**
53:                     $stop \leftarrow True$
54:         $timestep \leftarrow timestep + 1$
55:
56:         **if** $timestep \equiv 0$ (mod 25) **then**
57:             Sample new $g$ from the semantic search policy
58:
59:         **if** $subtask\_success$ **then**
60:             $goal\_pointer \leftarrow goal\_pointer + 1$
61:             $obj_i \leftarrow obj_{goal\_pointer}$; $action_i \leftarrow action_{goal\_pointer}$
62:             $move\_pointer \leftarrow 0$
63:             $execute\_interaction \leftarrow False$
64:             $stop \leftarrow False$
65:             $subtask\_success \leftarrow False$
66:             Sample new $g$ from the semantic search policy
67:             break
68:
69: **if** $goal\_pointer == k + 1$ **then**
70:     $task\_success \leftarrow True$

---

## A.6 MORE EXPLANATIONS ON TABLE 3

Table 3 shows common error modes and the percentage they take out of all failed episodes, with regards to SR. More specifically, it is showing the distribution of episodes into exactly one error mode, out of the 79.9% of all "Val Unseen" episodes that have failed (the episodes not in the 20.10% of Table 2). The common error modes are failures in (1) locating the subgoal object (due to the small field of view, imperfect segmentation, ineffective exploration), (2) locating the subgoal object because it is in a closed receptacle (cabinet, drawer, etc), (3) interaction (due to object being too far or not in field of view, bad segmentation mask), (4) navigation (collisions), (5) correctly processing language instructions, (6) others, such as the deterministic policy repeating a loop of actions from depth/ segmentation failures and 10 failed actions accruing from a mixture of different errors. These errors occur in the order of (5), (1)/ (2), (3), (4) in an episode, since the LP module operates in the beginning and the object has to be first localized to be interacted with, etc. If an episode ended with errors in multiple categories, it was classified as an example of an "earlier" error in making Table 3. For example, if the language processing module made an error and later there were also 10 collisions, this episode shown as a case of error (5) in Table 3.

## A.7 ASSIGNMENTS OF ROOMS INTO "LARGE" AND "SMALL" IN VALID UNSEEN

There are 4 distinct scenes in Valid Unseen (one kitchen scene, one living room, one bed room, one bathroom). The kitchen (Large) has a significantly larger area than all the others (Small).

## A.8 PROTOCOLS FOR REPRODUCING THE SEMANTIC POLICY

The primary result in Table 1 is from architecture tuning of the language processing, the semantic mapping, and the semantic search policy modules on the development data (validation unseen). Reviewers correctly noted that it is possible random seeds will also effect performance so the model was retrained four additional times and test results are reported here. Since components of the language processing and the semantic mapping module were trained from pre-trained weights, we report the performance of FILM with semantic search policy trained from different seeds.

The improvement by the semantic policy as shown in Table 1 is reproducible across multiple seeds. Table 8 shows results on Tests Unseen with semantic policy trained with different starting seeds (where SEED 1 denotes that the policy was trained with `torch.manual_seed(1)`). With learning rate of 0.001 and evaluation of every 50 steps, the model with the lowest test loss subject to train loss $< 0.62$ was chosen. The exact code and commands can be found here: https://github.com/soyeonm/FILM#train-the-semantic-policy.

**Table 8:** Results of FILM reproduced across different starting seeds of the semantic policy. The $\pm$ error bar in the AVG. row denotes the sample variance.

| Method | Tests Unseen | | | |
|---|---|---|---|---|
| | PLWGC | GC | PLWSR | SR |
| **Low-level + High-level Instructions** | | | | |
| TABLE 1 | 15.06 | 36.37 | 10.55 | 26.49 |
| SEED 1 | 15.12 | 38.55 | 11.34 | 27.86 |
| SEED 2 | 13.82 | 36.58 | 10.13 | 25.96 |
| SEED 3 | 10.47 | 37.12 | 14.05 | 25.64 |
| SEED 4 | 14.22 | 37.37 | 10.69 | 26.62 |
| AVG. | 13.74 | 37.20 | 11.352 | $26.51 \pm 0.58$ |
| **High-level Instruction Only** | | | | |
| TABLE 1 | 13.13 | 34.75 | 9.67 | 24.46 |
| SEED 1 | 14.05 | 36.75 | 10.47 | 25.51 |
| SEED 2 | 12.60 | 34.59 | 9.07 | 23.48 |
| SEED 3 | 12.86 | 35.02 | 9.23 | 23.68 |
| SEED 4 | 13.61 | 36.10 | 10.10 | 25.18 |
| AVG. | 13.25 | 35.44 | 9.71 | $24.87 \pm 0.64$ |

## A.9 A LANGUAGE PROCESSING MODULE WITHOUT THE TEMPLATE ASSUMPTION

The second paragraph of section 4.1 explains the template assumption, with the tasks belonging to one of the 7 types. For direct comparison with existing methods that do not take direct advantage of this assumption, we trained a new Language Processing module that does not make use of templates

but makes use of the subtasks sequences annotations ALFRED provides.[5] Fine-tuning a pre-trained BART (Lewis et al., 2020) model, we directly learned a mapping from a high-level instruction to a sequence of subtasks (e.g. "Drop a clean pan on the table" → "(PickupObject, Pan), (PutObject, Sink), ..."). Without any assumption on the structure of the input and the output, this model takes a sequence of tokens as input and outputs a sequence of tokens. With the new LP module, we obtained SR of 18.03% on valid unseen, which is a slight drop compared to our original 20.10%, indicating that templates are only marginally helpful in performance.

For future research, we believe templates should be used instead of subtasks annotations, since they are much cheaper to obtain in naturalistic settings. In this work, we created the 7 templates (one for each type) by writing down an intuitive canonical set of interactions to successfully perform the task. To do so, we looked at just 7 episodes in the training set and spent less than 20 minutes creating them; these cheaply obtained templates cover all 20,000 training episodes. Even to train an agent to perform more complex tasks, it is more realistic to use templates than assume sub-task annotations.

On the other hand, our findings simultaneously suggest the need for a better program synthesis method from instructions to subtask sequences, for general purpose instruction following not bound to certain "types" of instructions.

---

[5]Existing works(Blukis et al., 2021; Kim et al., 2021; Zhang & Chai, 2021; Pashevich et al., 2021) use subtask sequence annotations (or expert trajectories that contain the subtask annotations) as well.

