# OpenReview forum: "FILM: Following Instructions in Language with Modular Methods"
_ICLR.cc/2022/Conference — ICLR 2022 Poster_

### Official Review · Reviewer_dBxN · 2021-11-02

**Correctness:** 3
**Technical Novelty And Significance:** 3
**Empirical Novelty And Significance:** 4
**Recommendation:** 6
**Confidence:** 5

**Main Review:**

**Strengths**
+ The proposed approach leads to impressive (SoTA) results on the ALFRED benchmark without using expert trajectories or low-level instructions.
+ The analysis presented is very interesting. Specifically, the authors have done a commendable job in discussing the potential failure modes of the system. The analysis might prove to be potentially insightful for future researchers working on this problem.
+ Barring a few missing details / inconsistencies (refer to the remainder of the review), the paper is easy to follow and the main ideas have been presented in a clear and coherent manner.

**Weaknesses**
+ While the authors have demonstrated strong results on the ALFRED benchmark with their approach, I feel that the overall idea of using modular components to tackle long-range, target-driven navigation is not new and has been explored sufficiently in prior work (https://arxiv.org/abs/1907.02022, https://arxiv.org/abs/2110.02207, https://arxiv.org/abs/2007.00643, https://arxiv.org/abs/2007.00643, https://openreview.net/pdf?id=HklXn1BKDH, https://arxiv.org/abs/1809.00786). Having said that, I didn't find any particularly novel insight from the perspective of using waypoint-guided navigation via modular methods (the fact that semantic/spatial memories help long range navigation is not very interesting at this point).
+ While I do agree with the evidence for the claim that FILM generalizes well, however, there seems to be a huge gap between the performance of FILM and other competing approaches under the setting where low-level instructions are used in Tests-Seen (with FILM turning out to be inferior than the rest). This suggests that the proposed approach is not able to make use of the low-level instructions optimally. This is also evidenced from the trends seen in Tab. 2. Can the authors share their thoughts about this?
+ Within the Language Processing module, the authors use the oracle sub-task decomposition template for each question type and fill in the blanks with the output from the argument predictor. This seems to be privileged information that would greatly help modular methods. the authors must clarify whether all competing methods make use of this information and if not, then does it still remain a valid comparison?
+ The text (although overall well written), seems to be missing some critical details that might make a full comprehension of their approach problematic. Please see below for specific examples on missing information.

**Some questions/clarifications/suggestions**
+ Although not a critical issue at all (doesn't factor into the review), but I felt that the name of their approach FILM might get confused / overloaded with the visual reasoning model with the same name (https://arxiv.org/abs/1709.07871). Just thought that the authors should be aware of this as an FYI, if they are not already.
+ While going through the Introductory sections of the paper, I felt that a concrete definition (in terms of inputs, outputs and assumptions) of the Embodied Instruction Following (EIF) task as defined on the ALFRED benchmark is missing. Without that, it feels like the authors are talking about EIF in the generic sense of the term. Within that context, it becomes a little confusing when a comparison with VLN is made (because, again, speaking in general terms, VLN is also an instance of EIF).  The text later clarifies the exact task definition in the subsequent sections, but I felt that the definition should have come sooner.
+ The text refers to ALFRED as an environment at one place. I'd consider it as a benchmark (or, a dataset) rather than an environment.
+ The description of the argument prediction sub-module of the LP module talks about 4 kinds of arguments: “obj”, “recep”, “mrecep” and “sliced”. Whereas, in the figures and in the later sections, there is mention of “parent” being one of the argument types in place of “recep”. The authors should clarify the list of possible argument types and fix the inconsistencies.
+ The paper doesn't talk in detail about how the low-level instructions are used by their proposed modular approach. What parts of the existing pipeline are modified and how?
+ What do the authors mean when they say that the task is deemed successful within a tolerance of "10 failed actions”? Do these refer to the low-level actions (forward, left, right, …)? Or, are these high-level goals such as "slice bread” or both? Might be useful for readers who are not aware of the specifics of the ALFRED benchmark evaluation.
+ Is the aggregation over time steps in the Semantic Map Module done using something like simple averaging? Or, via some sequence models?
+ It seems like the number of semantic categories whose information is being saved at each of the MxM metric locations on the top-down map is different for different episodes (C+num of sub-goal objects). How is this handled?
+ Is the agent working with an egocentric or an allocentric map? The Figures seem to suggest an allocentric map. But the description of the semantic search policy states that the GT location for small objects is computed every 25 steps, which seems to suggest an egocentric map. The authors should consider making this explicit in the text.
+ How is the overall framework keeping track of completed sub-tasks and deciding to advance to the next one?

**Summary Of The Paper:**

The paper proposes a modular framework for solving instruction following tasks from the ALFRED benchmark. The approach uses a language module that takes the high-level instruction, predicts the instruction type and uses the oracle template for the instruction type to fill in the arguments (objects/receptacles) and generate a list of sub-tasks to be executed on those objects and receptacles. The agent builds and maintains a spatial+semantic map to keep track of state and uses it to plan the next waypoint (which object/receptacle to navigate to next) as its intermediate goal. Their approach demonstrates the ability to ground small objects on the top-down semantic map and yields SoTA performance on the ALFRED benchmark. The authors also present a comprehensive analysis of their ablations and failure modes.

**Summary Of The Review:**

I am looking forward to clarifications on missing information from the authors during the discussion phase.  Overall, while I do agree that the proposed approach shows impressive results on the ALFRED benchmark and the paper has useful insights, however, the overall approach doesn't have much novelty to offer. Therefore, I feel that the paper is good/useful, but slightly below the acceptance threshold due to novelty concerns.

---

> ### Author Response · Authors · 2021-11-19
> **Response to feedback from reviewer dBxN**
>
> We thank you for your valuable comments and feedback which will help us in revising the paper.
>
> For direct comparison with existing methods, we trained a new Language Processing module that does not make use of templates but makes use of the subtasks sequences annotations ALFRED provides. With the new LP module, we obtained SR of 18.03% on valid unseen*, which is a slight drop compared to our original 20.10%, indicating that templates are only marginally helpful in performance. However, this is still very high compared to 10.1% of Blukis et al [1]. For future research, we believe templates should be used instead of subtasks annotations, since they are much cheaper to obtain in naturalistic settings. Please see the general comment for more explanation.
>
> Regarding the novelty of the approach, while using modular components for navigation has been popular since the 1980’s ([2], [3], [4]) and only recently has seen re-emergence in VLN systems like those you referenced, unlike recent approaches, our work is on interaction with navigation where others have only worked on the simpler navigation task. This creates new challenges, since the interaction with and localization of small objects has not been touched upon much in the ``modular’’ approaches for navigation, additionally, the search space is also substantially larger due to state-changes and articulated objects. Furthermore, putting various components altogether to make a SOTA system is a novel attempt.
>
> Finally, our approach was designed to use high-level instructions only, as we believe this to be the most useful direction for future research. However, we provided results with low-level instructions (for which we simply concatenated to high-level instructions as input to the LP module) only for a more direct comparison with existing methods.
>
> For your suggestions, we have already incorporated some of them in the newly uploaded version, and we will also do so for the rest in the next revision. Again, thank you for your review!
>
> [1]: Blukis, V., Paxton, C., Fox, D., Garg, A., & Artzi, Y. (2021). A persistent spatial semantic representation for high-level natural language instruction execution. arXiv preprint arXiv:2107.05612.
>
> [2]: Ayache N, Faugeras OD. Building, Registrating, and Fusing Noisy Visual Maps. The International Journal of Robotics Research. 1988;7(6):45-65. doi:10.1177/027836498800700605
>
> [3]: J. L. Crowley, World modeling and position estimation for a mobile robot using ultrasonic ranging. Proceedings, 1989 International Conference on Robotics and Automation, 1989, pp. 674-680 vol.2, doi: 10.1109/ROBOT.1989.100062.
>
> [4]: Leonard J.J., Feder H.J.S. (2000) A Computationally Efficient Method for Large-Scale Concurrent Mapping and Localization. In: Hollerbach J.M., Koditschek D.E. (eds) Robotics Research. Springer, London. https://doi.org/10.1007/978-1-4471-0765-1_21

---

> > ### Author Response · Authors · 2021-11-22
> > **Clarification of the response**
> >
> > Dear reviewer dBxN,
> >
> > Here is some clarification for points that may have been ambiguous in the response.
> >
> > 1. The cited work from Blukis ([1]) uses the subtasks sequence annotations. The subtask sequence annotations(or expert trajectories that contain the subtask annotations) are also used in other existing works ([2], [3], [4]) as well. We will include the new results and make these points clear in the next revision.
> >
> > 2. The new LP model is a seq-to-seq model that does not involve any templates. Fine-tuning a pre-trained BART [5] model, we directly learned high-level instruction -> sequence of subtasks (e.g. “Drop a clean pan on the table” -> “(PickupObject, Pan), (PutObject, Sink), ...”). Thus, with the new LP module, we can directly use the output from the BART model with minimal post-processing.  The full code with training, post-processing, and predicted templates are here: https://drive.google.com/drive/folders/1PnB-yDAqy2KqQzH70NjnJr-ZuNt68I8b?usp=sharing
> >
> > [1]: Blukis, V., Paxton, C., Fox, D., Garg, A., & Artzi, Y. (2021). A persistent spatial semantic representation for high-level natural language instruction execution. arXiv preprint arXiv:2107.05612.
> >
> > [2]: Byeonghwi Kim, Suvaansh Bhambri, Kunal Pratap Singh, Roozbeh Mottaghi, and Jonghyun Choi.
> > Agent with the big picture: Perceiving surroundings for interactive instruction following. In
> > Embodied AI Workshop CVPR, 2021.
> >
> > [3]: Yichi Zhang and Joyce Chai. Hierarchical task learning from language instructions with unified
> > transformers and self-monitoring. arXiv preprint arXiv:2106.03427, 2021.
> >
> > [4]: Alexander Pashevich, Cordelia Schmid, and Chen Sun. Episodic transformer for vision-and language navigation. arXiv preprint arXiv:2105.06453, 2021.
> >
> > [5]: Lewis, Mike, et al. "Bart: Denoising sequence-to-sequence pre-training for natural language generation, translation, and comprehension." arXiv preprint arXiv:1910.13461 (2019).

---

### Official Review · Reviewer_qMNi · 2021-11-03

**Correctness:** 3
**Technical Novelty And Significance:** 2
**Empirical Novelty And Significance:** 2
**Recommendation:** 6
**Confidence:** 4

**Main Review:**

**Strengths**

- The presented method is currently (at the time of this review) the SOTA model based on the unseen test set split performance. This is the main strength I am basing my score on.
- The experimental setup presents a thorough set of ablations and error analysis (please see below for some questions).
- The semantic search policy, to my understanding, improves over the closely related work of Blukis et al. by replacing their random search policy.

**Weaknesses**

- In my perception the presented model is very close in structure to that of Blukis et al. 2021, with the main difference being that this model does not have a high-level controller and instead leverages domain knowledge to predict a templated high level plan. I believe assuming to know the task type taxonomy is a strong assumption to make and one that makes me concerned about the method's ability to generalize to other domains (particularly those for which such a taxonomy would not be made available). If accepted, I would encourage future versions of the paper to explicitly mention this set of assumptions. This point is the main weakness I am basing my score on.
- Minor weakness, but I felt like some of the figures were a little difficult to parse and certain experimental details appear to be missing (see below).
- Minor: The experiments presented in Table 6 are used to claim that the semantic search module helps particularly in larger scenes. However, the unseen validation set consists of only 4 scenes, each of a different category of room (kitchen, bathroom, living room, bedroom). I'm concerned that there may be a confounding factor in terms of the semantics of different rooms, and I'm wondering if any additional analysis was conducted to disentangle the effects of room size vs. that of room semantic category?

**Suggestions**

- Title: FILM is an acronym already in use by Perez et al. 2018 (FiLM: Visual Reasoning with a General Conditioning Layer), which may confuse readers. Just wanted to point this out for the author's consideration.
- Abstract: "This requires the use of expert trajectories and low-level language instructions" — this statement feels a little strong to me and I would suggest qualifying it since not all existing methods require low-level language instructions.
- Embodied Instruction Following: Very minor comment, but I have not seen this acronym be used before and I'm not sure if categorizing the task like this may confuse readers. Have other works used this term?
- Figure 1: (1) I found the red dots a bit confusing — it's not clear to me in which direction the agent is moving due to lack of arrows and them persisting across timesteps. (2) It took a while for me to understand what the labels on the semantic map mapped to, perhaps color coding them to match the semantic segmentation would make it clearer. (3) Why is the blue square so regular (ie a perfect square), whereas the rest of the semantic segmentation map is not?
- Page 6: "HLSM" was not previously defined in the text.
- Table 1: I would suggest stating what the bold, blue, and underlined numbers mean to make it easier to parse by readers.
- Figure 6 seems appears to be a little too close to Tables 6 and 7 — at first glance it appeared that "Instruction: Put a large clean knife on the counter" was supposed to be read with Table 6.
-

**Questions**

- How do you determine what objects are "small" and which are "large"? I was also confused by the statement in Section 2 comparing the semantic exploration policy to that of Chaplot et al.: "employs a semantic exploration policy, our and their semantic policies serve fundamentally..."
- If the model was run across multiple seeds, I'm wondering what the standard deviations were for the presented results?
- How were failure cases classified into each of the 6 error modes? I was very interested by this experiment, and I think the presentation could be made stronger if this is specified in the text.

**Summary Of The Paper:**

This paper presents FILM (Following Instructions in Language with Modular methods), a model built for the ALFRED dataset. FILM consists of:

1) A language processing module which uses a BERT backbone to (a) predict 1 out of 7 task types in the ALFRED ontology and (b) predicts slot categories for populating a template (1 pre-defined per task).

2) A semantic mapping module which uses segmentation and depth prediction to populate a 2D spatial map of object class locations.

3) A semantic search policy which uses a CNN to map from the semantic map to subgoal object locations.

4) A deterministic policy which is used to navigate to either (a) a subgoal object location if it exists in the semantic map or (b) a sampled location from the search policy if not, and performs a predefined object interaction based on the current subgoal from the task template.

In my perception the main technical contribution of the work comes in the form of the language processing module, which provides much of the structure used for planning actions by the agent.

The paper also contributes a semantic search policy for reasoning over a semantic map to predict subgoal object locations (instead of randomly picking a new search location).

Experimentally, the paper presents:

1) SOTA results on the task when only the high-level instruction is given (low-level instructions witheld).

2) Ablations showing a bottleneck from errors in the perceptual side and marginal gains from adding low-level language instructions.

3) A taxonomy and distribution of error modes for the presented model, showing most errors come from either the language prediction, goal object location prediction, or goal objects being in closed receptacles (which the deterministic policy does not open).

4)  An analysis showing that task type length does not correlate with performance.

**Summary Of The Review:**

I am leaning in favor of acceptance because the method presented achieves SOTA performance on the ALFRED task, showing how domain knowledge can be leveraged for improved performance.

However, I do not feel strongly about this stance because the performance boost appears to be achieved by baking in domain knowledge that may keep the method from being generalizable to other domains.

Despite this concern, I think this work would still be of interest to the community since it may also serve to highlight the need for improvements in high-level planning/control in these types of visual language navigation tasks. Similarly, other components of the model are also hand-crafted and could serve to highlight ripe areas of work for the community (e.g. a low-level deterministic policy is used, which does not open receptacles).

---

> ### Author Response · Authors · 2021-11-19
> **Response to feedback from reviewer qMNi**
>
> We thank you for your valuable comments and feedback which will help us in revising the paper.
>
> To address the usage of templates, we trained a new Language Processing module that does not make use of templates but makes use of the subtasks sequences annotations ALFRED provides. With the new LP module, we obtained SR of 18.03% on valid unseen, which is a slight drop compared to our original 20.10%, indicating that templates are only marginally helpful in performance. However, this is still very high compared to 10.1% of Blukis et al [1]. For future research, we believe templates should be used instead of subtasks annotations, since they are much cheaper to obtain in naturalistic settings. Please see the general comment for more explanation.
>
> To address confounding variables, we agree that there must be some confounding between the size of the room and its semantics. In the next revision, we will examine the performance of each modular component (semantic segmentation, language processing, depth prediction, semantic search) across room types and incorporate our observations. We have been working on retraining (above) and have not been able to do this analysis yet.
>
> For your questions, we deemed ``obj`` and ``recep``, defined in ALFRED, are "small" and "large" objects, since the first kind can be picked up with one hand. In FILM, the semantic policy is dependent on "seeds" (other components were fine-tuned from pre-trained weights such as ``bert-base-uncased``, ``maskrcnn_resnet50_fpn``); in the next revision, we will report results with the semantic policy trained across multiple seeds. We have added explanations on the division of failure cases in the new revision (Section 5.2, Appendix A.6).
>
> For your suggestions, we have already incorporated some of them in the newly uploaded version, and we will also do so for the rest in the next revision. Again, thank you for your review!
>
> [1]: Blukis, V., Paxton, C., Fox, D., Garg, A., & Artzi, Y. (2021). A persistent spatial semantic representation for high-level natural language instruction execution. arXiv preprint arXiv:2107.05612.

---

> > ### Author Response · Authors · 2021-11-22
> > **Clarification of the response**
> >
> > Dear reviewer qMNi,
> >
> > Here is some clarification for points that may have been ambiguous in the response.
> >
> > 1. The cited work from Blukis ([1]) uses the subtasks sequence annotations. The subtask sequence annotations(or expert trajectories that contain the subtask annotations) are also used in other existing works ([2], [3], [4]) as well. We will include the new results and make these points clear in the next revision.
> >
> > 2. The new LP model is a seq-to-seq model that does not involve any templates. Fine-tuning a pre-trained BART [5] model, we directly learned high-level instruction -> sequence of subtasks (e.g. “Drop a clean pan on the table” -> “(PickupObject, Pan), (PutObject, Sink), ...”). Thus, with the new LP module, we can directly use the output from the BART model with minimal post-processing.  The full code with training, post-processing, and predicted templates are here: https://drive.google.com/drive/folders/1PnB-yDAqy2KqQzH70NjnJr-ZuNt68I8b?usp=sharing
> >
> > [1]: Blukis, V., Paxton, C., Fox, D., Garg, A., & Artzi, Y. (2021). A persistent spatial semantic representation for high-level natural language instruction execution. arXiv preprint arXiv:2107.05612.
> >
> > [2]: Byeonghwi Kim, Suvaansh Bhambri, Kunal Pratap Singh, Roozbeh Mottaghi, and Jonghyun Choi.
> > Agent with the big picture: Perceiving surroundings for interactive instruction following. In
> > Embodied AI Workshop CVPR, 2021.
> >
> > [3]: Yichi Zhang and Joyce Chai. Hierarchical task learning from language instructions with unified
> > transformers and self-monitoring. arXiv preprint arXiv:2106.03427, 2021.
> >
> > [4]: Alexander Pashevich, Cordelia Schmid, and Chen Sun. Episodic transformer for vision-and language navigation. arXiv preprint arXiv:2105.06453, 2021.
> >
> > [5]: Lewis, Mike, et al. "Bart: Denoising sequence-to-sequence pre-training for natural language generation, translation, and comprehension." arXiv preprint arXiv:1910.13461 (2019).

---

> > > ### Comment · Reviewer_qMNi · 2021-11-30
> > > **Response**
> > >
> > > Thanks to the authors for their thoughtful response and subsequent modifications based on the feedback!
> > >
> > > In summary, I still stand by my original assessment and believe that the ideas/experiments provided in this paper would be of interest to the research community. I will maintain my score in favor of acceptance with the proposed modifications.
> > >
> > > Regarding the usage of templates, I still retain the same concerns, which have also been expressed by other reviewers. While I agree that it has been clearly shown here that using templates is helpful in ALFRED, I don't believe this paradigm would scale to truly natural domains (ie. not simulations). In the real world, the same subtask sequence may not always work given the same instruction (e.g. say for example a case where the countertop is cluttered with objects and must first be cleaned, or the cabinet door is jammed, etc.). The templates used here are based on the data-generating process itself, so I think it's intuitive that they improve performance.
> > >
> > > With that said, I think the usage of templates here is an assumption that the method makes, and the paper clearly shows improvements on the benchmark under that assumption. Independent of perspective on this, I believe that this result itself is interesting and could contribute to future work since it might help isolate other components of existing systems which may be further improved.

---

> > > > ### Author Response · Authors · 2021-12-03
> > > > **World dynamics and limitations**
> > > >
> > > > Hi, thanks again for engaging with the work!
> > > >
> > > > I think this point is really interesting and brings up a section we will include in the final version of the paper about the relationship between scripts/templates and world dynamics.  I think another angle through which to view your comment is "If you know the world dynamics, you can define precise scripts -- if you don't, they must be underspecified in key ways" where I'm using scripts here to reference the script learning literature.  Script learning, is notoriously difficult because the goal is to build high-level plans that are flexible and elide details (i.e. "go to the airport" without specifying the mode of transportation).
> > > >
> > > > There is no (near future) robotic platform with truly open-ended manipulation/control so it will be the case (even in the cluttered scene) that there are restricted dynamics which can be specified with high-level scripts but I think what you're actually pointing out is that such scripts will require failure+recovery modes.  In our setting, there are failure models that the scripts do not capture (e.g. due to perception, maximum number of collisions, etc) and one's that are implicit to the architecture (e.g. search).
> > > >
> > > > Through this lens, a future paper which includes, for example, Sim2Real experiments would need to create a system for handing physics failures etc but I do not think it would fundamentally change the high-level TAMP formulation, only some of the underlying modular components and/or replanning.
> > > >
> > > > I would love to continue this discussion and try to actually drill down into where you think changes would need to be made for a physical platform that don't fall under this umbrella, but we have probably exhausted the ICLR forum functionality so I'll stop here and hope to continue in some other future forum.
> > > >
> > > > Thanks again!

---

### Official Review · Reviewer_NHgG · 2021-11-03

**Correctness:** 3
**Technical Novelty And Significance:** 3
**Empirical Novelty And Significance:** 3
**Recommendation:** 6
**Confidence:** 4

**Main Review:**

# Thank you

Thank you for your comments. After reading them and other reviews, I find myself convinced in part by the arguments. In particular this one on the templates. I see the authors commented about them. I think this account but an important part of the generalization.
>"FILM makes use of 7 highly specialized and ALFRED-specific task templates. This takes advantage of the templated nature of the tasks in the ALFRED environment, somewhat trivializes the language understanding task, and will not generalize to new tasks without new template engineering."
However, it's fairly common in robotics tasks to have a series of high-level plans.

I also became more concerned about novelty.

I'm revising down my score. Thanks again.

# Previous main review
The main contribution of the paper is the combination of the modules.
1. Parsing the instruction. Purely based on NLP
2. Exploring the map for building the global map and to locate objects. Based on a combination of SLAM and estimation of object presence.
3. Interleaving the exploration with the selection of the next tasks/goal necessary for achieving the goal of the instructions.
4. Using a hand-written policy for going to the next point of exploration.

Module 2 is also a contribution per se, as it is designed to address the difficulty of finding small objects. Finally, modules 3 and 4 are contributions, not as isolated innovation but as they show their empirical power.

Besides the impressive difference in performance, I am convinced about the argument of modularity. For instance, from natural language input, only the goal instruction is found to be necessary. Improvement in NL processing would independently increase the quality of the approach. Such separation might prevent interpreting the instructions given what was found in the environment. That's not relevant for ALFRED as a limited kind of object appears in all the domains.

In my opinion, the paper has two main potential weak aspects, but I don't think they are ground for rejection.
- The granularity of the semantic search policy might be hard to control. However, I am convinced by the discussion in appendix A.4
- The deterministic policy is not just using FMM. The pseudo-code in the appendix also includes a policy for exploring the current location.

I have the following question I'd like the authors to answer:

- I'm confused about how the arguments are used. Section 4.1 explains the 4 types. Later on, "recep" is easier to detect while "obj" is harder.
	- Are these four kind of arguments treated differently by the semantic mapping module?
	- C is the number of "objects", but I am not sure if that refers to the 1st kind of argument or objects in general.
	- On the other hand, the policy is trained only with the receptacle. What about the "sliced" and "mrecep"?
	- Please provide a summary of how the different kind of arguments are used across the whole paper.
	- Please indicate where such clarification would be added in the paper.

Here are other comments and questions that I don't think deserve a direct answer during the rebuttal:
- Table 3: is this regarding the overall performance shown in the 1st row of table 4? Are these errors with respect to SR or GC? I think it'd be more illuminating to present this table in absolute numbers. One option would be to add the row of no errors so that the columns add up to 100.
- page 6. Evaluation Splits. Would you please remind the reader the size of those sets? That would help to evaluate the significance of the metrics.
- Table 2: It's not clear that "base method" is FILM with semantic search. I only confirmed it because "base method" is the same as "overall" in table 4.
	- Page 8: "Thus, we use FILM with semantic search as the "base method" (default) for all experiments/ ablations". Would you please say this before introducing Table 2?
- page 8: "Second, the number of the subtasks entailed with the task type does not strongly correlate with performance." This is not surprising since the templates make the length less important. I think it'd be better to mention the effective number of combinations or suggest that as real proxy for difficulty.

**Summary Of The Paper:**

The paper presents a modular approach for solving Vision Language Navigation tasks and test it in ALFRED, a recent benchmark for that task. Compared with other benchmarks, ALFRED features a higher number of actions, objects, and an agent operating with an egoistic view. Solving the task requires multiple high-level actions, combined with low-level actions and exploration. As suggested by the authors of ALFRED, the benchmark requires exploiting the hierarchical structure, reusing skills in multiple contexts, and structured reasoning. The submission uses precisely all these aspects to achieve a significant improvement over the next entry in the ALFRED leaderboard.

**Summary Of The Review:**

In summary, the paper proposes a modular approach for Vision Language Navigation tasks. The assumptions are heavily customized for ALFRED, but some general lessons can be used to other domains.

The most interesting contribution was the level of abstraction of the semantic map and policy that allows separating the exploration and exploitation from the natural language instructions and from acting given the current situation and information in the map. Furthermore, given that actions are related to objects, and the agent is located in an environment, it makes sense to have a predictable policy, allowing to adjust independently the issues related to the map.

So, I propose acceptance.

---

> ### Author Response · Authors · 2021-11-19
> **Response to feedback from reviewer NHgG**
>
> We thank you for your valuable comments and feedback which will help us in revising the paper.
>
> For your questions, an object in ALFRED is either a ``recep`` or ``obj`` (or non-receptacle) object. ``parent`` objects (which was mistakenly written as ``mrecep`` in some parts of the original version) are special kinds of ``recep`` objects. ``sliced’’ is not a kind of object but it indicates whether an object should be sliced before being picked up.
>
> Across the whole paper, the 4 kinds of arguments are used as follows. The policy is trained with the ``recep`` objects as input and predicts locations of ``obj`` objects. ``sliced`` and ``parent`` are only used in the language processing module. In the semantic mapping module, $C$ refers to the objects in general (all ``recep`` objects and ``obj`` (equivalently, ``non-receptacle``) objects of interest in the current episode). We agree that the distinction of ``obj`` and ``object``  should be more clear to the reader. We have incorporated these contents in Section 4.2 and Appendix A.2 in the newly uploaded version (edits are in purple). Please let us know if you think there are still some unclear or missing explanations.
>
> For your suggestions, we have already incorporated some of them in the newly uploaded version, and we will also do so for the rest in the next revision. Again, thank you for your review!

---

> ### Author Response · Authors · 2021-12-02
> **Apology and Update to Reviewer NHgG**
>
> Dear Reviewer NHgG,
>
> We realized we did not include this detail in our response to you and thank you for your early support of our paper. For your concern on templates, please note that we trained a new Language Processing module that does not make use of templates but makes use of the subtasks sequences annotations ALFRED provides (which are also used by previous works such as [1,2,3,4]). Fine-tuning a pre-trained BART [5] model, we directly learned high-level instruction -> sequence of subtasks (e.g. “Drop a clean pan on the table” -> “(PickupObject, Pan), (PutObject, Sink), ...”). The full code with training, post-processing, and predicted templates are here: https://drive.google.com/drive/folders/1PnB-yDAqy2KqQzH70NjnJr-ZuNt68I8b?usp=sharing
>
> With the new LP module, we obtained SR of 18.03% on valid unseen*, which is a slight drop compared to our original 20.10%, indicating that templates are only marginally helpful in performance. However, this is still very high compared to 10.1% of Blukis et al [1].
> For future research, we believe templates should be used instead of subtasks annotations, since they are much cheaper to obtain in naturalistic settings.
>
> [1]: Blukis, V., Paxton, C., Fox, D., Garg, A., & Artzi, Y. (2021). A persistent spatial semantic representation for high-level natural language instruction execution. arXiv preprint arXiv:2107.05612.
>
> [2]: Byeonghwi Kim, Suvaansh Bhambri, Kunal Pratap Singh, Roozbeh Mottaghi, and Jonghyun Choi. Agent with the big picture: Perceiving surroundings for interactive instruction following. In Embodied AI Workshop CVPR, 2021.
>
> [3]: Yichi Zhang and Joyce Chai. Hierarchical task learning from language instructions with unified transformers and self-monitoring. arXiv preprint arXiv:2106.03427, 2021.
>
> [4]: Alexander Pashevich, Cordelia Schmid, and Chen Sun. Episodic transformer for vision-and language navigation. arXiv preprint arXiv:2105.06453, 2021.
>
> [5]: Lewis, Mike, et al. "Bart: Denoising sequence-to-sequence pre-training for natural language generation, translation, and comprehension." arXiv preprint arXiv:1910.13461 (2019).

---

### Official Review · Reviewer_aTek · 2021-11-05

**Correctness:** 2
**Technical Novelty And Significance:** 3
**Empirical Novelty And Significance:** 2
**Recommendation:** 6
**Confidence:** 3

**Main Review:**

This paper presents a modular system (FILM) for egocentric instruction following in the ALFRED environment. The system does not require expert trajectories and can operate without low-level instruction sequences. However, it does make use of templated mappings from 7 high-level goal types to low-level instruction sequences, and these are quite specific to the tasks in ALFRED. As well as presenting a novel language understanding module that uses engineered instruction-sequence templates, the paper introduces a new semantic search strategy that identifies potential locations for each of the small object sub-goals, based on large receptacle object locations. The semantic map building module, and within sub-goal deterministic policy are motivated by previous work. However their incorporation into the FILM system is novel.

Experiments show that FILM generalizes well from environments that were seen during training to those that weren't. This generalization is impressive and in contrast to previous systems, which all suffer from a significant performance drop when evaluated on unseen environments. FILM is evaluated both in the setting where low-level instructions sequences are proivided, and when only high-level goals are available. FILM performs better than previous work on unseen environments across the board; worse than previous work on with low-level instructions on seen instructions; and marginally better than previous work without low-level instructions on seen environments. However, the comparison to other work that doesn't use low-level instructions is a bit murky. FILM has access to a small set of task templates that somewhat trivialize the high-level to low-level mapping and I think this needs to be discussed more.

**Strengths**
- FILM generalizes much better to unseen rooms than previous work.
- Semantic search policy improves accuracy, efficiency, and is particularly effective in large rooms and with difficult to detect sub-goals. This bodes well for even more complex environments.
- NLP module mapping high-level natural-language instructions to low-level instruction sequences is very effective and maintains most of the performance of the system that has access to low-level instructions. Outperforms other systems that don't have access to low-level instructions.

**Weaknesses**
- The comparison to other systems that don't have access to low-level instructions is a bit murky. FILM makes use of 7 highly specialized and ALFRED-specific task templates. This takes advantage of the templated nature of the tasks in the ALFRED environment, somewhat trivializes the language understanding task, and will not generalize to new tasks without new template engineering. I feel the approach is valid but the paper would be stronger with a more thorough discussion of the tradeoffs between sytems that require low-level instruction sequences; systems that are constrained to a predefined set of templated instruction sequences; and systems that learn the high-level to low-level mapping without constraints.


**Summary Of The Paper:**

This paper presents a modular system (FILM) for egocentric instruction following in the ALFRED environment. The system does not require expert trajectories and can operate without low-level instruction sequences. The system has the following components:

1. A language processing module that maps high-level instructions "Drop a clean pan on the table" to a low-level sub-task sequence. This module makes use of a number of BERT based classifiers that:

   a.  classifiy the high level instruction into one of seven possible goal types, each of which is deterministically associated with a templated sub-task sequence

   b. identify object IDs to fill in the slots in the template identified in (a)

2. A semantic mapping module that populates a grid with binary classifications that indicate object presence; obstacle presence; and whether the location has been explored. This module is inspired by previous work on egocentric navigation.

3. A novel semantic search policy that predicts the likely location of small sub-goal objects (bowl) on the basis of large receptacle objects (table). This is trained via supervised learning on sampled trajectories and it samples goals at a coarse tiem scale of every 25 steps.

4. A deterministic policy that operates within each 25 step search goal period. This makes use of the Fast Marching Method and is inspired by previous work on this task.

Together, these modules make up a system that generalizes to unseen environments much better than previous work. The new semantic search policy provide small but consistent improvements over a version of FILM that has this module removed.

FILM is evaluated in both the setting where low level instruction sequences are provided, and the setting where only the high-level goal is provided and FILM must map this goal to low-level instructions via module (1). In both settings, FILM significantly outpeforms previous work on unseen environments. When low-level instructions are provided, FILM does worse than some previous work on seen environments. When low-level instructions aren't provided, FILM does marginally better than previous work on seen environments.

Experiments show that the semantic search policy increases path-weighted metrics proportionally more than absolute metrics, suggesting that this policy improves efficiency as well as overall success of the search. Targeted evaluations also show that this search policy is particularly useful in large rooms, where the agent cannot see many target objects, and for the "Clean and Place" task that involves a difficult to detect object (sink) contained in a simple to detect object (countertop).

**Summary Of The Review:**

- Novel modular egocentric instruction following system with multiple technical contributions.
- Generalization to unseen environments is impressive.
- Dependence on task-specific instruction templates needs to be discussed in relation to previous work. Claim that this can be directly compared to other work that does not make use of low-level instructions is questionable.

---

> ### Author Response · Authors · 2021-11-19
> **Response to feedback from reviewer aTek**
>
> We thank you for your valuable comments and feedback which will help us in revising the paper.
>
> For direct comparison with existing methods, we trained a new Language Processing module that does not make use of templates but makes use of the subtasks sequences annotations ALFRED provides. With the new LP module, we obtained SR of 18.03% on valid unseen, which is a slight drop compared to our original 20.10%, indicating that templates are only marginally helpful in performance. However, this is still very high compared to 10.1% of Blukis et al [1].
>
> For future research, we believe templates should be used instead of subtasks annotations, since they are much cheaper to obtain in naturalistic settings. In our case, the first author created the 7 templates (one for each type) by writing down an intuitive canonical set of interactions to successfully perform the task. To do so, we looked at just 7 episodes in the training set and spent less than 20 minutes creating them; these cheaply obtained templates cover all 20,000 training episodes. Even to train an agent to perform more complex tasks, it is more realistic to use templates than assume sub-task annotations.
>
> We have already incorporated some of your suggestions in the newly uploaded version, and we will also do so for the rest in the next revision. Again, thank you for your review!
>
> [1]: Blukis, V., Paxton, C., Fox, D., Garg, A., & Artzi, Y. (2021). A persistent spatial semantic representation for high-level natural language instruction execution. arXiv preprint arXiv:2107.05612.

---

> > ### Comment · Reviewer_aTek · 2021-11-21
> > **Clarification on use of subtask sequence annotations.**
> >
> > Thank you for this update. For a little more context, could you please also clarify:
> >
> > 1. Whether the cited work from Blukis et.al. also uses the subtask sequence annotations.
> > 2. A little about how the new LP module works. It is unclear to me how the BERT-based classification approach would extend beyond a templated representation of the tasks.

---

> > > ### Author Response · Authors · 2021-11-22
> > > **Response with clarification**
> > >
> > > Thank you very much for your comment!
> > >
> > > 1. The cited work from Blukis ([1]) uses the subtasks sequence annotations. The subtask sequence annotations(or expert trajectories that contain the subtask annotations) are also used in other existing works ([2], [3], [4]) as well. We will include the new results and make these points clear in the next revision.
> > >
> > > 2. The new LP model is a seq-to-seq model that does not involve any templates. Fine-tuning a pre-trained BART [5] model, we directly learned high-level instruction -> sequence of subtasks (e.g. “Drop a clean pan on the table” -> “(PickupObject, Pan), (PutObject, Sink), ...”). Thus, with the new LP module, we can directly use the output from the BART model with minimal post-processing.  The full code with training, post-processing, and predicted templates are here: https://drive.google.com/drive/folders/1PnB-yDAqy2KqQzH70NjnJr-ZuNt68I8b?usp=sharing
> > >
> > > [1]: Blukis, V., Paxton, C., Fox, D., Garg, A., & Artzi, Y. (2021). A persistent spatial semantic representation for high-level natural language instruction execution. arXiv preprint arXiv:2107.05612.
> > >
> > > [2]: Byeonghwi Kim, Suvaansh Bhambri, Kunal Pratap Singh, Roozbeh Mottaghi, and Jonghyun Choi.
> > > Agent with the big picture: Perceiving surroundings for interactive instruction following. In
> > > Embodied AI Workshop CVPR, 2021.
> > >
> > > [3]: Yichi Zhang and Joyce Chai. Hierarchical task learning from language instructions with unified
> > > transformers and self-monitoring. arXiv preprint arXiv:2106.03427, 2021.
> > >
> > > [4]: Alexander Pashevich, Cordelia Schmid, and Chen Sun. Episodic transformer for vision-and language navigation. arXiv preprint arXiv:2105.06453, 2021.
> > >
> > > [5]: Lewis, Mike, et al. "Bart: Denoising sequence-to-sequence pre-training for natural language generation, translation, and comprehension." arXiv preprint arXiv:1910.13461 (2019).

---

### Author Response · Authors · 2021-11-19
**General comment on the role of templates**

We would like to thank all the reviewers for their insightful and valuable comments. Here is a general comment on our usage of templates.

For direct comparison with existing methods, we trained a new Language Processing module that does not make use of templates but makes use of the subtasks sequences annotations ALFRED provides. Fine-tuning a pre-trained BART [5] model, we directly learned high-level instruction -> sequence of subtasks (e.g. “Drop a clean pan on the table” -> “(PickupObject, Pan), (PutObject, Sink), ...”). The full code with training, post-processing, and predicted templates are here: https://drive.google.com/drive/folders/1PnB-yDAqy2KqQzH70NjnJr-ZuNt68I8b?usp=sharing. With the new LP module, we obtained SR of 18.03% on valid unseen, which is a slight drop compared to our original 20.10%, indicating that templates are only marginally helpful in performance. However, this is still very high compared to 10.1% of Blukis et al [1].

With the new LP module, we obtained SR of 18.03% on valid unseen*, which is a slight drop compared to our original 20.10%, indicating that templates are only marginally helpful in performance. However, this is still very high compared to 10.1% of Blukis et al [1]. For future research, we believe templates should be used instead of subtasks annotations, since they are much cheaper to obtain in naturalistic settings.


For future research, we believe templates should be used instead of subtasks annotations, since they are much cheaper to obtain in naturalistic settings. In our case, the first author created the 7 templates (one for each type) by writing down an intuitive canonical set of interactions to successfully perform the task. To do so, we looked at just 7 episodes in the training set and spent less than 20 minutes creating them; these cheaply obtained templates cover all 20,000 training episodes. Even to train an agent to perform more complex tasks, it is more realistic to use templates than assume sub-task annotations.

[1]: Blukis, V., Paxton, C., Fox, D., Garg, A., & Artzi, Y. (2021). A persistent spatial semantic representation for high-level natural language instruction execution. arXiv preprint arXiv:2107.05612.

Thank you once again for your valuable feedback!

---

### Public Comment · ~Mingyu_Ding1 · 2021-11-20
**Nice work! Do you have plan to release the code?**

Dear Authors,

Thank you for your great work, which provides a lot of insights to the community!

I'm wondering if you have a plan to release the codes and models? It will definitely help a lot for further following your work. Thank you!

All the best.

---

> ### Author Response · Authors · 2021-11-20
> **Absolutely!**
>
> We plan to make our GitHub public immediately after the anonymity period -- and will look into a mechanism for sharing code anonymously sooner.  Thanks for your interest, we'll try and update this thread as soon as we have a solution!

---

### Public Comment · ~So_Yeon_Min2 · 2022-02-21
**Code released!**

The official github of FILM is now released here:
https://github.com/soyeonm/FILM

This code will serve as the starting code for the ALFRED challenge held at the CVPR2022 EAI 2022 Workshop!
https://askforalfred.com/EAI21/

---

### Decision · Program_Chairs · 2022-01-20

**Decision:**

Accept (Poster)

**Comment:**

This paper develops a modular system named FILM, for egocentric instruction execution task in the ALFRED environment, which uses structured representations that build a semantic map of the scene, perform exploration with a semantic search policy, to achieve the natural language goal. They achieve strong performance while avoiding both expert trajectories and low-level instructions. The reviewers all reasonably liked the paper (all reviewers gave 'marginally above the acceptance threshold' score) and appreciated the planner ideas + strong results; but many of them also had concerns about the use of templated mappings from 7 high-level goal types to low-level instruction sequences, and whether this will make the system specific to ALFRED. The authors did provide some new results in the response period to show that results drop without the templates but not by a large margin. Some reviewers also had concerns about the novelty of the work and said that the semantic map building module and sub-goal deterministic policy are motivated by previous work, but their incorporation into the FILM system is novel. Lastly, there was some concerns/debates on whether the system assumes/uses too much domain knowledge / task type taxonomy which might reduce the ability to generalize to other domains / data types, versus on the other hand the results may also serve to highlight the need for improvements in high-level planning/control in these types of visual language navigation tasks.